High cytoplasmic YAP1 expression predicts a poor prognosis in patients with colorectal cancer

Dong Tianqi 1
Yuan Yuncang 1
Xiang Xudong 2
Sang Shuping 1
Shen Hao 1
Wang Lei 1
Yang Chunyan 1
Li Fangfang 1
Li Hongliang lihongliang@ynu.edu.cn 1
Zheng Shangyong shangyong@ynu.edu.cn 1
1 School of Medicine, Yunnan University , Kunming , Yunnan , China
2 Department of Thoracic Surgery, Third Affiliated Hospital of Kunming Medical University , Kunming , Yunnan , China
Albertini Maria Cristina
Electronic publication date: 2020 Nov 19
Publication date: 2020
Volume: 8
Electronic Location ID: e10397
Received 2020 Jun 15; Accepted 2020 Oct 29
Copyright: ©2020 Dong et al.
Copyright year: 2020
Copyright holder: Dong et al.
License: This is an open access article distributed under the terms of the Creative Commons Attribution License, which permits unrestricted use, distribution, reproduction and adaptation in any medium and for any purpose provided that it is properly attributed. For attribution, the original author(s), title, publication source (PeerJ) and either DOI or URL of the article must be cited.
License URL: https://creativecommons.org/licenses/by/4.0/

Keywords: Colorectal cancer, YAP1, Subcellular localization, Immunohistochemistry, Prognosis, Chemotherapy

Funding: The National Natural Science Foundation of P. R. China 81460435 81860494 This work was supported by the grants from the National Natural Science Foundation of P. R. China (No. 81460435 and No. 81860494 to Shangyong Zheng). The funders had no role in study design, data collection and analysis, decision to publish, or preparation of the manuscript.

==============================
Purpose

Yes associated protein 1 (YAP1), which is a standout amongst the most essential effectors of the Hippo pathway, assumes a vital part in a few kinds of cancer. However, whether YAP1 is an oncogene in CRC (colorectal cancer) remains controversial, and the association between the subcellular localization of YAP1 and clinical implications in CRC remains unknown.

Patients and methods

In this study, we investigated the subcellular localization of YAP1 in CRC cells by immunohistochemistry and then associate these findings with clinical information in a large CRC cohort with 919 CRC patients.

Results

The results show that CRC tissues has a significant higher expression of cytoplasmic YAP1 compared to adjacent normal tissues (all P < 0.001). Cytoplasmic YAP1 expression was significantly associated with the number of lymph nodes removed and differentiation grade (all P < 0.001). Furthermore, after correcting confounding variables, for example, TNM stage and differentiation grade, the multivariate Cox analysis confirmed cytoplasmic YAP1-high subgroup had a significant shorter DFS (HR = 3.255; 95% CI [2.290–4.627]; P < 0.001) and DSS (HR = 4.049; 95% CI [2.400–6.830]; P < 0.001) than cytoplasmic YAP1-low subgroup. High cytoplasmic YAP1 expression is associated with a worse survival in stage III CRC patients who received chemotherapy.

Conclusion

Cytoplasmic YAP1 could be could be utilized as a prognosis factor in CRC patients, and may be an indicator of whether certain patients population could benefit from postoperative chemotherapy.

Introduction

The prevalence of CRC ranks third among all cancers in male and second in female (Torre et al., 2015), and CRC has the third highest mortality rate among all cancers (Zeng et al., 2014). Surgical resection combined with chemotherapy remains the mainstay of treatment for CRC, in any case, numerous patients will progress to metastatic CRC and develop resistance to chemotherapeutic drugs (Fisher et al., 2015), because signs or symptoms diagnose CRC usually appear in advanced phases (Binefa et al., 2014). Even if some patients are diagnosed with CRC and undergo surgery at an early stage, 20% –30% of these patients will relapse within five years. (Hardingham et al., 2015). The current CRC treatment regimen is heterogenous for patients, even for patients with the same TNM stage (Nagtegaal, Quirke & Schmoll, 2012); in any case, the indication for treatment should be assessed on an individual basis by considering the risk factors of relapse (Marin et al., 2012). Currently, the only effective marker for the CRC prognosis and appropriate chemotherapy selection is microsatellite instability (MSI) (Hemminki et al., 2000; Popat, Hubner & Houlston, 2005); however, MSI as a CRC marker has not been applied clinically. Therefore, there is an urgent need for new biomarkers to assess the prognosis of CRC patients before and after treatment.

YAP1 is a standout amongst the most essential effectors of the Hippo pathway, which is a critical pathway regulating cell proliferation, apoptosis, and organ growth (Justice et al., 1995). Several studies have shown that YAP1 is an oncogene highly express in numerous cancer types including bladder cancer (Liu et al., 2013), breast cancer (Kim, Jung & Koo, 2014), gastric cancer (Kang et al., 2011), hepatocellular cancer (Xu et al., 2009), nonsmall-cell lung cancer (Wang et al., 2010), and CRC (Wang et al., 2013a; Xu et al., 2009) that associate with tumor progression and poor prognosis. On the contrary, abundant literature suggested that YAP1 is a tumor suppressor gene and nuclear expression is reduced in different cancers, such as breast cancer (Matallanas et al., 2007; Yu et al., 2013; Yuan et al., 2008), head and neck cancers (Ehsanian et al., 2010), hematological cancers (Cottini et al., 2014), and CRC (Levy et al., 2007). These paradoxical reports remind us that the role of YAP1 in cancer is controversial, and it is crucial to make it clear the relationship between YAP1 expression and its clinical relevance in CRC. In addition, the nuclear overexpression of YAP1 is associate with poor survival in gastric cancer (Kang et al., 2011), actually, previous researches suggest that subcellular localization of proteins is associated with functions associated of tumorigenesis and tumor progression (Garcia et al., 2005; Lobo et al., 2009; Vaquero et al., 2017), a few studies suggested that YAP1 overexpression is associate with poor survival in CRC (Wang et al., 2013a; Wang et al., 2013b; Yang et al., 2018); however, the association between subcellular localization of YAP1 and clinical significance in CRC has been largely ignored. Thus, the prognostic significance of YAP1 in CRC needs further investigation.

On the basis of these considerations, we explore the the subcellular localization of YAP1 in 929 CRC colorectal tissues by immunohistochemistry performed on tissue microarrays (TMAs), and investigate the association between subcellular localization of YAP1 and patient’s survival. This study may provide deeper insights to understand the role of YAP1 in prognosis and treatment of CRC.

Materials and Methods

Bioinformatics analysis

All available related mRNA expression profiles of YAP1 in CRC tissues were downloaded from the TCGA database and normalized by EDASeq package which can take gene length and GC-content into account. The microarray expression profiles of eight datasets (GSE8671, GSE37364, GSE41258, GSE23878, GSE22598, GSE9348, GSE81582, and GSE77955) associated with CRC tissues were randomly downloaded from the Gene Expression Omnibus (GEO) database, then the mRNA expression profiles of YAP1 in different pathological feature CRC tissues were extracted from the microarray expression profiles and compared by independent sample t-tests or paired sample t-tests.

Patient characteristics

Between January 2001 and November 2011, we collected a total of 1067 CRC tissue samples donated by 929 patients who underwent surgery at Yunnan Cancer Hospital&Third Affiliated Hospital of Kunming Medical University (Kunming, P. R. China), then seven tissue microarrays (TMAs) were constructed with these samples by a biotechnology company (Outdo Biotech, Shanghai, P. R. China) as reported previously (Pan et al., 2015). The core on the TMA is 1.2 mm in diameter and one core represents a sample, and there are 70 normal samples, 33 adenoma samples, 949 primary cancer samples, and 15 metastatic cancer samples on TMAs. Pathological diagnosis and staging of all patients based on the 7th American Joint Committee on Cancer Staging System, the clinic characteristics of all patients, including age, sex, disease location, TNM stage, differentiation grade, number of resected lymph nodes, chemotherapy (FOLFOX regimen), serum carcinoembryonic antigen (CEA), and carbohydrate antigen 19-9 (CA19-9) were summarized in Table 1. This study was approved by The Committee on Human Subject Research and Ethics, Yunnan University (approval number: yuncare20200358). All patients signed a written informed consent for using their tissues for research purpose.

Table 1 Associations of cytoplasmic YAP1 expression with demographic and clinical variables of 919 CRC patients.

Characteristics	Total(n = 919)	CytoplasmicYAP1expression level	Pvalue*	
		Low(n = 457)	High(n = 462)		
Mean age ± SD(year)	60.1 ± 12.4	61.4 ± 12.3	60.7 ± 12.5	0.389**	
Sex (n (%))				0.686	
Men	549(59.7)	270(59.1)	279(60.4)		
Women	370(40.3)	187(40.9)	183(39.6)		
Disease location(n(%))				0.632	
Rectum	512(55.7)	251(54.9)	261(56.5)		
Colon	407(44.3)	206(45.1)	201(43.5)		
Differentiation grade(n(%))				<0.001***	
Well	95(10.3)	67(14.7)	28(6.1)		
Moderately	779(84.8)	369(80.7)	410(88.7)		
Poorly	35(3.8)	14(3.1)	21(4.5)		
Missing	10(1.1)	7(1.5)	3(0.6)		
Resected lymph nodes (n(%))				<0.001	
<12	201(21.9)	140(30.6)	61(13.2)		
≥12	718(78.1)	317(69.4)	401(86.8)		
TNM stage (n(%))				0.362***	
I	140(15.2)	65(14.2)	75(16.2)		
II	459(49.9)	245(53.6)	214(46.3)		
III	320(34.8)	147(32.2)	173(37.4)		
Chemotherapy(n(%))				0.730	
Yes	671(73.0)	336(73.5)	335(72.5)		
No	248(27.0)	121(26.5)	127(27.5)		
Serum CEA (n(%))				0.451	
<5 ng/ml	568(61.8)	288(63)	280(60.6)		
≥5 ng/ml	351(38.2)	169(37)	182(39.4)		
Serum CA19-9 (n(%))				0.686	
<37U/ml	788(85.7)	394(86.2)	394(85.3)		
≥37U/ml	131(14.3)	63(13.8)	68(14.7)		
Notes.

* χ2 test.

** Student t-test.

*** Mann–Whitney U test (non-parametric). Missing values are excluded for all statistic tests.

Abbreviations YAP1 Yes associated protein 1

TNM tumor-node-metastasis

CEA carcinoembryonic antigen

CA19-9 carbohydrate antigen 19-9

Immunohistochemistry (IHC)

IHC is performed on 4 µm thick array slides. Specifically, the array slides were primarily immersed into the citrate solution (pH 6.0) and boil for 5 min for antigen retrieval, then incubated with 10% goat serum (SL038; Solarbio, Beijing, P. R. China) for 30 min at room temperature to block non-specific binding, subsequently, Mouse antihuman YAP1 primary monoclonal antibody (1:100, sc-376830; Santa Cruz Biotechnology, Santa Cruz, CA, USA) was used to incubate array slides at 4 °C overnight, and secondary antibody included in the Maxvision™ HRP-Polymer Anti-Mouse IHC Kit (KIT-5920; Maxvision, P. R. China) was used to incubate array slides 10 mins at room temperature. All array slides performed IHC simultaneously and strictly comply with the standard protocol.

Quantitative evaluation of immunostaining

Aperio ScanScope (Aperio Technologies, Vista, CA, USA) was used to digitally scan the stained TMA slides, then the scan image can be used for Quantitative evaluation of immunostaining, the YAP1 protein expression level was quantified by H-score method as reported previously (Detre, Jotti & Dowsett, 1995). Specifically, the staining intensity in the epithelial cell was scored as 0, 1, 2, or 3 corresponding to the presence of negative, weak, intermediate, and strong brown staining, respectively, then the number of cells stained at each intensity was counted. The H-score is the multiplication of the proportion of positive cells and the corresponding staining intensity score (0, 1, 2 or 3), thus an H-score between 0 and 300 was obtained. The quantitative evaluation of immunostaining was performed separately by two co-authors who were blinded to the clinicopathological information, and the scores were averaged.

Follow-up and patients

The follow-up information for 919 CRC patients was collected using a standard methods reported previously (Pan et al., 2015). Disease-free survival (DFS) was defined as the number of months from the first treatment to the first relapse. Disease-specific survival (DSS) as the number of months from the first treatment to the date of death due to CRC. The patients were divided into two subgroups (cytoplasmic YAP1 high vs. cytoplasmic YAP1 low, nuclear YAP1 high vs. nuclear YAP1 low, or YAP1 NCR high vs. YAP1 NCR low) by the optimal cut-off values for maximum discrimination in survival difference, the cut-off values were determined by the maxstat R package in R 3.2.0.

Statistical analysis

Clinicopathological characteristics of all CRC patients were summarized in related tables, in which continuous variables were tested by two-sample Student t-tests, and categorical variables were tested by Pearson Chi-squared tests, the TNM stage and differentiation grade were test by Wilcoxon-Mann–Whitney tests. The DSS and DFS of patient’s subgroups were compared by Kaplan–Meier analysis with log-rank test to examine the difference. All factors were determined their independence of the prognostic value by Univariate and multivariate Cox regression analyses (Further evaluation of meaningful prognostic factors in univariate analysis in multivariate analysis). All statistical analyses were conducted by SPSS 21 for Windows (IBM Inc., Armonk, NY, USA) and it was considered statistically significant if P < 0.05.

Results

Differences in YAP1 expression between CRC tissues and adjacent normal tissues.

To analyze the expression pattern of YAP1 in CRC tissues, we firstly utilized the datasets from public database, the results showed that in one TCGA dataset and three GEO datasets, YAP1 mRNA expression level was consistently significantly elevated in CRC tissues compared with the adjacent normal tissue (all P < 0.01; Figs. 1A–1D), the other five GEO datasets also show the same results (all P < 0.01; Fig. S1A). We subsequently investigated the expression pattern of YAP1 by IHC method in 997 CRC and 70 adjacent normal tissue samples which derive from patients who underwent surgery at Yunnan Cancer Hospital. The positive immunostaining results from YAP1 predominantly occurred in the cytoplasm and nucleus of colorectal epithelial cells (Figs. 1H–1M), whereas the staining was negative or weak in mesenchymal cells (Figs. 1H–1M). We calculated the H-score of cytoplasmic YAP1 and nuclear YAP1 independently, then the YAP1 NCR (nuclear/cytoplasmic ratio) was calculated, and there is a very weak positive correlation between cytoplasmic H-score and nuclear YAP1 H-score (Fig. S1B). the results show that cytoplasmic YAP1 expression was significantly elevated in CRC tissues compared with the adjacent normal tissues (all P < 0.001; Fig. 1E), and nuclear YAP1 expression was significantly elevated in primary cancer tissues compared with the adjacent normal tissues (P < 0.0001; Fig. 1F), but the expression of nuclear YAP1 in adenomas and metastasis CRC tissues have no significant differences with the adjacent normal tissues (Fig. 1F), we also found YAP1 NCR (Nuclear/Cytoplasmic Ratio) gradually decrease in adjacent normal tissues, adenomas, primary cancers, and metastatic CRC (all P < 0.05; Fig. 1G). The results above indicated that the increased cytoplasmic YAP1 expression may be associated with the progression of CRC.

Figure 1 Differences in YAP1 expression between CRC tissues and adjacent normal tissues.

(A–D) Bioinformatics analyses of YAP1 mRNA expression between cancer and cancer related specimens in one TCGA dataset and three GEO datasets. (E) Comparison of YAP1 expression level among different colorectal pathological tissues by cytoplasmic YAP1 H-score, (F) nuclear YAP1 H-score, or (G) YAP1 NCR H-score. (H–M) Representative YAP1 staining in normal tissues and cancer tissues, the blue staining represents the nuclear staining and the brown staining represents the YAP1 positive staining, cancer tissue have the higher cytoplasmic YAP1 H-score, higher nuclear YAP1 H-score and lower NCR than normal tissue, scale bars: 100 µm. * P < 0.05; ** P < 0.01; *** P < 0.001; **** P < 0.0001; ns, no significance.

Associations between YAP1 expression and CRC patients’ clinicopathological characteristics

To obtain further information, we analyzed the association between cytoplasmic YAP1 expression levels or YAP1 NCR and CRC patients’ clinicopathological characteristics. We found that the expression of cytoplasmic YAP1 protein was significantly higher in poorly+moderate grades than that in the well grade (P < 0.001; Fig. S1C), and YAP1 NCR was significantly lower in poorly+moderate grades than that in the well grade (P < 0.001; Fig. S1E), but there is no significant differences between poorly+moderate grades and well grade in the expression of nuclear YAP1 protein (Fig. S1D), the clinicopathological features for the patients at poor+moderate grade or well grade were described in Table S1. Next, we classified the 919 patients (patients lost follow-up information were excluded) into cytoplasmic YAP1-low and cytoplasmic YAP1-high subgroups by the optimal cut-off value (H-score = 202.5) determined by the maxstat R package, the results showed there were significant differences between cytoplasmic YAP1-low and cytoplasmic YAP1-high subgroups with respect to the number of resected lymph nodes and differentiation grade (all P < 0.001; Table 1). We also classified the 919 patients into YAP1 NCR-low and YAP1 NCR-high subgroups by the optimal cut-off value (NCR = 0.0482) determined by the maxstat R package, the results showed a significant difference between the YAP1 NCR-low and YAP1 NCR-high subgroups in the TNM stage (P = 0.02; Table S2). The above results revealed that high cytoplasmic YAP1 expression may be involved in the aggressiveness of CRC.

High cytoplasmic YAP1 expression is associated with a worse survival in CRC patients

A univariate and multivariate Cox regression analyses (Further evaluation of meaningful prognostic factors in univariate analysis in multivariate analysis) was applied to determined the independence of the prognostic value of YAP1 on the basis of the DFS and DSS of CRC patients, the results showed that high cytoplasmic YAP1 expression was an independent risk factor of DFS (HR = 3.255; 95% CI [2.290–4.627]; P < 0.001) and DSS (HR = 4.049; 95% CI [2.400–6.830]; P < 0.001) for CRC patients (Table 2), likewise, low YAP1 NCR was an independent risk factor of DFS (HR = 2.295; 95% CI [1.118–4.711]; P = 0.024) and DSS (HR = 2.873; 95% CI [1.045–7.902]; P = 0.041) for CRC patients (Table 2), but the univariate Cox regression analysis showed that nuclear YAP1 expression level was not a meaningful prognostic factor either for DFS (HR = 0.684; 95% CI [0.453–1.031]; P = 0.07) or DSS (HR = 0.860; 95% CI [0.412–1.975]; P = 0.688) for CRC patients (Table 2). Kaplan–Meier analyses with log-rank tests showed that DFS and DSS in the cytoplasmic YAP1-high subgroup were significantly shorter than the cytoplasmic YAP1-low subgroup (all P < 0.001; Figs. 2A, 2E), moreover, cytoplasmic YAP1-high subgroups were consistently had shorter DFS and DSS than cytoplasmic YAP1-low subgroups in stage I, II, or III CRC patients respectively (all P < 0.01; Figs. 2B–2D, 2F–2H). We also found DFS and DSS were significantly lower in YAP1 NCR-low subgroup than YAP1 NCR-high subgroup (all P < 0.01; Fig. S2A). However, there is no significant differences between nuclear YAP1-high subgroup and nuclear YAP1-low subgroup in Kaplan–Meier analyses (all P > 0.05; Fig. S2B).

Table 2 Cox regression analysis of immunohistochemistry YAP1 expression and clinicopathological covariates in patients with CRC.

Characteristics	Disease-free Survival		Disease-specific Survival	
	Univariate		Multivariate		Univariate		Multivariate	
	HR (95%CI)	P Value		HR (95%CI)	P Value		HR (95%CI)	P Value		HR (95%CI)	P Value	
YAP1-high vs. YAP1-low(cytoplasmic)	3.891 (2.758–5.490)	<0.001		3.255 (2.290–4.627)	<0.001		4.291 (2.545–7.236)	<0.001		4.049 (2.400–6.830)	<0.001	
YAP1-low vs. YAP1-high(NCR)	2.709(1.331–5.511)	0.006		2.295(1.118–4.711)	0.024		3.346(1.219–9.183)	0.019		2.873(1.045–7.902)	0.041	
YAP1-high vs. YAP1-low(nuclear)	0.684(0.453–1.031)	0.070					0.860(0.412–1.975)	0.688				
Age (>=60 vs. <60)	0.897 (0.667–1.207)	0.474					0.891 (0.568–1.398)	0.617				
Sex (female vs. male)	0.867 (0.638–1.177)	0.360					0.817 (0.512–1.302)	0.395				
Location (colon vs. rectum)	1.068 (0.793–1.440)	0.665					1.159 (0.737–1.823)	0.522				
TNM (per increase in stage)	1.874 (1.474–2.381)	<0.001		1.863 (1.471–2.360)	<0.001		1.256 (0.889–1.775)	0.196				
Grade (per increase in grade)	3.001 (1.948–4.625)	<0.001		3.435 (2.127–5.548)	<0.001		2.992 (1.575–5.685)	0.001		2.732 (1.383–5.394)	0.004	
Chemotherapy (yes vs. no)	1.705 (1.156–2.515)	0.007		1.029 (0.647–1.637)	0.902		1.125 (0.662–1.912)	0.663				
Resected lymph nodes (≥12 vs. <12)	2.675 (1.689–4.236)	<0.001		1.780 (1.111–2.853)	0.017		2.610 (1.375–4.954)	0.003		1.685 (0.874–3.251)	0.120	
Serum CEA (≥5 vs. <5 ng/ml)	1.646 (1.223–2.215)	0.001		1.513 (1.120–2.043)	0.007		1.378 (0.876–2.170)	0.166				
Serum CA19-9 (≥37 vs. <37 U/ml)	1.766 (1.224–2.549)	0.002		1.350 (0.914–1.995)	0.132		1.619 (0.906–2.894)	0.104				
Notes.

Abbreviations HR hazard ratio

CI confidence interval

YAP1 Yes associated protein 1

TNM tumor-node-metastasis

CEA carcinoembryonic antigen

CA19-9 carbohydrate antigen 19-9

NCR Nuclear/Cytoplasmic Ratio

Figure 2 High cytoplasmic YAP1 expression is associated with worse survival in CRC patients.

(A–D) Associations between cytoplasmic YAP1 expression and DFS in the patient subgroups with different stage. (E–H) Associations between cytoplasmic YAP1 expression and DSS in the patient subgroups with different stage. Patients with stages I–III, stage I, stage II, or stage III were dichotomized into the cytoplasmic YAP1-high subgroups and cytoplasmic YAP1-low subgroups according to optimal cut-off value. Kaplan–Meier survival curves reveal DFS and DSS in patients with each TNM stage CRC. P-values are from Kaplan-Meier analysis with log-rank test.

High cytoplasmic YAP1 expression is associated with a worse survival in stage III CRC patients who received chemotherapy

To evaluate whether cytoplasmic YAP1 expression level could be an indicator of whether certain patients population could benefit from adjuvant chemotherapy, the stage III patients were divided into two groups respectively (all stage III patients received adjuvant chemotherapy), either did or did not receive adjuvant chemotherapy (Table S3), for stage III patients who received adjuvant chemotherapy, Kaplan–Meier analyses with log-rank tests showed that DFS and DSS in the cytoplasmic YAP1-high subgroup were significantly shorter than the cytoplasmic YAP1-low subgroup (all P < 0.001; Figs. 3A–3B), but there were no significant differences between YAP1-high subgroup and YAP1-low subgroup in DFS and DSS for stage III patients without adjuvant chemotherapy (all P > 0.05; Figs. 3C–3D). Besides, for stage III patients who received adjuvant chemotherapy, Kaplan–Meier analyses also showed that DFS and DSS in the low YAP1 NCR subgroup were significantly shorter than the high YAP1 NCR subgroup (all P < 0.001; Figs. S3A, S3B), but there were no significant differences between high YAP1 NCR subgroup and low YAP1 NCR subgroup in DFS and DSS for stage III patients without adjuvant chemotherapy (all P > 0.05; Figs. S3C–S3D). Therefore, high cytoplasmic YAP1 expression is associated with a worse survival in stage III CRC patients who received chemotherapy.

Figure 3 High cytoplasmic YAP1 expression is associated with a worse survival in stage III CRC patients who received chemotherapy.

Associations between cytoplasmic YAP1 expression and DFS (A) or DSS (B) in the stage III patients with chemotherapy. Associations between cytoplasmic YAP1 expression and DFS (C) or DSS (D) in the stage III patients without chemotherapy. P-values are from Kaplan-Meier analysis with log-rank test.

Discussion

Primarily, the results in this study showed that in one TCGA dataset and eight GEO datasets, the mRNA expression of YAP1 in CRC tissues was consistently higher in CRC tissues compared with the adjacent normal tissue. Further, the IHC examination of YAP1 confirmed that epithelial cytoplasmic YAP1 protein expression were significantly elevated in CRC tissues compared with the adjacent normal tissue in the Yunnan Cancer Hospital, and YAP1 NCR gradually decrease in adjacent normal tissues, adenomas, primary cancers, and metastatic CRC. Prior studies had illustrated the expression of YAP1 in a various cancers including CRC (Cottini et al., 2014; Ehsanian et al., 2010; Kang et al., 2011; Kim, Jung & Koo, 2014; Levy et al., 2007; Liu et al., 2013; Matallanas et al., 2007; Wang et al., 2013a; Wang et al., 2010; Xu et al., 2009; Yu et al., 2013; Yuan et al., 2008), but the association between subcellular localization of YAP1 and aggressiveness of CRC has been neglected. In this study, the expression pattern of YAP1 in the Yunnan Cancer Hospital cohort reveal that the increased cytoplasmic YAP1 expression may be associated with the progression of CRC.

The analysis of association between YAP1 expression and CRC patients’ clinicopathological features showed that cytoplasmic YAP1 expression was related to differentiation grade and YAP1 NCR was related to TNM stage. Further, CRC patients were divided into cytoplasmic-high YAP1 and cytoplasmic-low YAP1 subgroups by the optimal cut-off value (H-score=202.5), meanwhile, classify CRC patients into YAP1 NCR-low and YAP1 NCR-high subgroups according to the optimal cut-off value (NCR=0.0482). We found that DFS and DSS in the cytoplasmic YAP1-high subgroup were significantly shorter than the cytoplasmic YAP1-low subgroup, and DFS and DSS were significantly lower in YAP1 NCR-low subgroup than YAP1 NCR-high subgroup. High cytoplasmic YAP1 expression and low YAP1 NCR were found to be independent risk factors for CRC prognosis in multivariate Cox analysis (after correcting confounding variables), above results indicated that cytoplasmic YAP1 may be used as an indicator for staging of tumor. This is the first study to show the potential association between subcellular localization of YAP1 and CRC patients’ clinicopathological characteristics.

Adjuvant chemotherapy (FOLFOX/CapeOX regimen) is currently the most effective cytotoxic regimen for the treatment of CRC, FOLFOX adjuvant therapy can significant improve the survival of CRC patients (Gustavsson et al., 2015). However, adjuvant chemotherapy also has some side effects, such as myelotoxicity, neurotoxicity or gastrointestinal toxicity which can be fatal and cause complications (Mohelnikova-Duchonova, Melichar & Soucek, 2014), therefore, biomarkers predicting the benefit of chemotherapy are urgently needed. Our study clearly demonstrated that high cytoplasmic YAP1 expression is associated with a worse survival in stage III CRC patients who received chemotherapy. Currently, microsatellite instability (MSI) is the only effective indicator for prognosis and suitable chemotherapy regime for colorectal cancer patients (Hemminki et al., 2000; Popat, Hubner & Houlston, 2005), therefore, a new biomarker is urgently needed to instruct us to determine if a population is suitable for adjuvant chemotherapy. Therefore, cytoplasmic YAP1 may have crucial clinical implications and deserve further study.

There is some evidence to suggest that YAP1 is retained in the cytoplasm by AKT phosphorylation (Basu et al., 2003) or through binding LATS1 (Matallanas et al., 2007), and YAP1 functions as an oncogene which can promote CRC progression by activating the ERK/PI3K-AKT signaling pathway (Wang et al., 2017; Zhang et al., 2016). And LATS1/2 has been reported have a suppress role in cancer immunity (Moroishi et al., 2016), and this phenomenon may be a reason why YAP1 cytoplasmic localization is associate with the progression and poor prognosis of CRC. In another way, YAP1 acts as a tumor suppressor gene interacting with p73 to cause transcription of proapoptotic gene puma (Matallanas et al., 2007), but the apoptosis can be suppressed by enhancing the retention of YAP1 in cytoplasm. This may be the reason why high cytoplasmic YAP1 expression and low YAP1 NCR is associated with the progression and poor prognosis of CRC. Recent research has suggested that upregulation of EGFR by YAP1 has contributed to confer chemoresistance to esophageal cancer cells (Song et al., 2015) , another study suggested that YAP1 confers Colon cancer cells chemoresistance to 5FU chemotherapy (Touil et al., 2014), Therefore, YAP1 may promote CRC progression, high cytoplasmic YAP1 expression is associated with a worse survival in stage III CRC patients who received chemotherapy. However, the suggestions above are speculative, further mechanistic studies are required to explain the results.

Conclusion

In this study, we provided important evidence that increased cytoplasmic YAP1 correlated with the malignant phenotype in CRC. Importantly, the results show that increased cytoplasmic YAP1 was significantly associated with poor prognosis in CRC patients. More importantly, high cytoplasmic YAP1 expression is associated with a worse survival in stage III CRC patients who received chemotherapy. Our study has revealed that Cytoplasmic YAP1 could be utilized as prognostic factors in CRC patients and may be indicators of whether a certain patient population could benefit from postoperative chemotherapy, however, the molecular mechanisms behind it remain unknown and need to be further investigated.

Supplemental Information

Supplemental Information 1 Supplementary bioinformatics analyses and differences in YAP1 expression between different differentiation grades

(A) Bioinformatics analyses of YAP1 mRNA expression levels in cancer and cancer-related specimens in five GEO datasets. (B) Correlation scatter plot of Cytoplasmic vs Nuclear YAP1 H-score. (C) Comparison of YAP1 expression levels between different differentiation grades by cytoplasmic YAP1 H-score (D) nuclear YAP1 H-score, or (E) YAP1 NCR H-score. * P < 0.05; ** P < 0.01; *** P < 0.001; **** P < 0.0001; ns, no significance.

Click here for additional data file.

Supplemental Information 2 Low YAP1 NCR is associated with a worse survival in CRC patients

(A) Associations between nuclear YAP1 expression and DFS or DSS in patients with stages -. Patients with stages - were dichotomized into nuclear YAP1-high subgroups and nuclear YAP1-low subgroups according to the optimal cut-off value. (B) Associations between the YAP1 NCR and DFS or DSS in patients with stages -. Patients with stages - were dichotomized into YAP1 NCR-high subgroups and YAP1 NCR-low subgroups according to the optimal cut-off value. The P-values are from Kaplan-Meier analysis with the log-rank test.

Click here for additional data file.

Supplemental Information 3 Adjuvant chemotherapy has a differential effect on patients with different YAP1 NCRs

Associations between the YAP1 NCR and DFS (A) or DSS (B) in stage patients with chemotherapy. Associations between the YAP1 NCR and DFS (C) or DSS (D) in stage patients without chemotherapy. The P-values are from Kaplan-Meier analysis with the log-rank test.

Click here for additional data file.

Supplemental Information 4 The clinicopathological features for the patients at poor+moderate grade or well grade

Notes:* χ 2 test. ** Mann–Whitney U test (non-parametric). Missing values are excluded for all statistic tests. Abbreviations: CEA, carcinoembryonic antigen; CA19-9, carbohydrate antigen 19-9.

Click here for additional data file.

Supplemental Information 5 Associations of YAP1 NCR with demographic and clinical variables of 919 CRC patients

Notes:* χ 2 test. ** Student t-test. *** Mann–Whitney U test (non-parametric). Missing values are excluded for all statistic tests. Abbreviations: YAP1, Yes associated protein 1; TNM, tumor-node-metastasis; CEA, carcinoembryonic antigen; CA19-9, carbohydrate antigen 19-9; NCR, Nuclear/Cytoplasmic Ratio.

Click here for additional data file.

Supplemental Information 6 The clinicopathological features for the patients at stage III with or without chemotherapy

Notes:* χ 2 test. ** Mann–Whitney U test (non-parametric). Missing values are excluded for all statistic tests. Abbreviations: CEA, carcinoembryonic antigen; CA19-9, carbohydrate antigen 19-9.

Click here for additional data file.

Supplemental Information 7 The TCGA and GEO datasets, survival ananlysis data conducted by SPSS 21, & whole field figure of representive IHC staining figure

Click here for additional data file.

Additional Information and Declarations

Competing Interests

Author Contributions

Human Ethics

Data Availability

The authors declare there are no competing interests.

Tianqi Dong conceived and designed the experiments, performed the experiments, analyzed the data, prepared figures and/or tables, authored or reviewed drafts of the paper, and approved the final draft.

Yuncang Yuan performed the experiments, analyzed the data, prepared figures and/or tables, and approved the final draft.

Xudong Xiang, Shuping Sang, Hao Shen, Lei Wang, Chunyan Yang and Li Fangfang performed the experiments, prepared figures and/or tables, and approved the final draft.

Hongliang Li and Shangyong Zheng conceived and designed the experiments, performed the experiments, authored or reviewed drafts of the paper, and approved the final draft.

The following information was supplied relating to ethical approvals (i.e., approving body and any reference numbers):

This study was approved by The Committee on Human Subject Research and Ethics, Yunnan University (approval number: yuncare20200358).

The following information was supplied regarding data availability:

The raw data are available in the Supplementary File.

The microarray expression profiles are available in eight NCBI GEO datasets: GSE8671, GSE37364, GSE41258, GSE23878, GSE22598, GSE9348, GSE81582, and GSE77955.

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
