# Peer review of "High cytoplasmic YAP1 expression predicts a poor prognosis in patients with colorectal cancer"

_PeerJ, doi:10.7717/peerj.10397_

## Round 0.1 · original submission · Major Revisions

The authors can address reviewers' comments.

Reviewer 1 ·

Basic reporting

The manuscript broadly meets all the criteria and expected standards. However, I recommend authors to provided a English-translated version of the written consent form (currently in Chinese), as this has been presented as a Supplemental evidence.

Experimental design

No Comments

Validity of the findings

I have a few comments on the results presented and conclusions drawn by the authors.

1. There is an instance of minor image duplication in Figure 1E. In the upper panel comprising of Normal (Upper) Cytoplasmic (Left) and Nuclear YAP1 (middle), authors use slightly shifted field-of-views of the same image to extract the inset crop. If possible, provide an image of another field for the Nuclear YAP1 data. If you intend to use the same image, please provide the exact same field of view for both data set and mention the same in the figure legend.

2. Authors report overall similar trends in their results when scoring for either ‘High cytoplasmic YAP1’ or ‘Low YAP1 NCR’. While these two are intuitively linked, the ratio could be biased by differences in Nuclear Levels of YAP1. Authors have provided independent quantification of the same in Fig 1 B-D. I feel this result could be bolstered by adding a correlation scatter plot of Cytoplasmic v/s Nuclear YAP1 H-score, to rule there are any non-trivial relations between them, which might bias their results.

Additional comments

The differential effects of chemotherapy on high cytosolic YAP1 cells constitute the key finding of this manuscript. Please highlight this in the abstract more clearly.

Reviewer 2 ·

Basic reporting

1. Clear legend should be provided for Figure 1E. It is not clear what the colors represents, and it is not clear what are the differences between the images presented and what is the message each image is delivering to the readers.
2. Comparing the H-scores in Figure 1E with that in Figure 1B-D, it seems that images presented in Figure 1E are examples of extreme H-scores. Those images are not representative staining.
3. In Line 62 of the Introduction, the authors mentioned the need of new biomarkers for CRC prognosis. I think the authors should introduce if there are other biomarkers for CRC prognosis and discuss about how YAP1 is different from or better than existing biomarkers.
4. In Lines 66-74. The authors first introduced the paradoxical reports of YAP1 either as an oncogene or a tumor suppressor gene, then proposed the need of establishing a clear relationship between YAP1 expression and its clinical relevance. I think the authors should discuss why they think the relationship they have established is clearer than those in previous reports instead of simply one more example being added to the existing paradoxical reports.
5. Raw data. A lot of information, including TNM stage, differentiation grade, manual grading of H-score before averaging, etc., are not provided in the raw data used for Figures 2-3 and Tables 1-2. Raw data for Figure 1B-E are not provided.
6. The font size in Figure 2 is too small.

Experimental design

1. In Lines 88-95 bioinformatics analysis, the authors used publicly available gene datasets. The authors should describe how and why they select those specific datasets for their analysis. Are those all the related datasets available from the database, or are they randomly selected from the database? In Line 90, the authors should provide more details on how the data are normalized.

Validity of the findings

1. In Lines 193-194, the authors concluded that high cytoplasmic YAP1 expression may contribute to CRC aggressiveness. This conclusion is not justified as the data only suggest for a correlation but not for a causation. It is equally likely that YAP1 expression is modified as CRC progresses to a more advanced stage, i.e. CRC progression contributes to cytoplasmic YAP1 expression.
2. The conclusion (Lines 215-216, Lines 228-229, and Lines 264-265) generated from Figure 3 is not justified. Firstly, the sample size for the groups without chemotherapy is too small, it is premature to draw any conclusions based on the current data (i.e. Fig. 3C-D, Fig. S3C-D). Secondly and most importantly, to evaluate whether chemotherapy has an effect on cytoplasmic YAP1-low group or cytoplasmic YAP1-high group, and whether the effect of chemotherapy is different due to the expression level of cytoplasmic YAP1 level, the authors should compare the group with chemotherapy (either high cytoplasmic YAP1 or low cytoplasmic YAP1) to the corresponding group without chemotherapy first, and then check, for example, if chemotherapy is effective for the cytoplasmic YAP1-low patients but not effective for the cytoplasmic YAP1-high patients. The current comparison in Figure 3 only answers the question whether chemotherapy influences the prognostic role of cytoplasmic YAP1 level, but not the question whether cytoplasmic YAP1 level influences the chemotherapy effect.

Additional comments

In this manuscript, Dong and colleagues investigated the relationship between YAP1 expression and colorectal cancer (CRC) prognosis. They found a correlation between CRC progression with cytoplasmic but not nuclear YAP1 expression, with high cytoplasmic YAP1 corresponding to poor prognosis. This is an interesting finding as YAP is a transcriptional co-activator which regulates gene expression when transported to the nucleus. However, some of the conclusions are not justified. Please see specific comments for details.

Reviewer 3 ·

Basic reporting

The authors have investigated the subcellular loacalization of YAP-1 protein in CRC samples and examined its potential correlation to CRC prognosis and Disease specific/free survival. They calculated YAP-1 staining intensity h scores and studies TMA.

The language is simple, clear and non ambiguous. Authors have provided proper references. A little descriptive background about why YAP-1 plays a role in CRC, its dual role in CRC as an oncogene and a tumor suppressor, and its association to hippo pathway might be insightful.

Experimental design

The overall observational analysis suggesting that higher nuclear and cytoplasmic YAP-1 expression is associated with aggravated CRC.
1. The authors suggest that YAP1 mRNA expression was elevated in 3 GEO and 1 TCGA dataset upon analysis following which they mention that other 5 GEO datasets also exhibit similar results. Is there a significance to segregating the first 3 and other 5 GEO datasets while analysis and result interpretation?
2. Suppression of YAP-1 phosphorylation leads to its higher nuclear accumulation and has been previously reported to be correlated to CRC prognosis. However, the authors have shown that CRC samples had both high nuclear and cytoplasmic YAP-1 expression. What is the significance of higher cytoplasmic YAP-1? YAP-1 regulates and drives multiple pathways and It would add to the study if authors look into the pathways that drive its high cytoplasmic expression in CRC.
3. Authors observe that subjects that underwent adjuvant chemotherapy and had high cytoplasmic YAP-1 had a shorter DFS and DSS. Authors can extrapolate this observation to in vitro studies to understand the actual mechanism of how cytoplasmic YAP-1 expression can determine the efficacy of adjuvant chemotherapy. If authors are able to verify this observation through mechanistic evaluation, it is the most important and interesting finding of this study.
4. Did authors see any age dependent or gender dependent cytoplasmic/ nuclear YAP-1 variability?
5. In the discussion section, authors have reported previous literature on how the upregulation of EGFR by YAP1 confers chemoresistance to esophageal cancer cells . It will be intersting to see if their observation on effect of high cytoplasmic YAP-1 on adjuvant chemotherapy is also due to EGF-R upregulation or other factors.
6. If I understand correctly, the cytoplasmic YAP-1 expression was high in all malignant CRC irrespective of stage. What is the prognostic value of YAP-1 in differentiating the stage based on its expression.

Validity of the findings

The study is interesting in many facets owing to its clinical significance however it is mostly observational and bioinformatic. To assess and confirm these observations, elucidating the underlying cause of YAP-1 cytoplasmic up-regulation in CRC will be highly useful.
The english needs minor revision and language editing. Overall the paper is easy to understand and direct. Fine tuning the discussion can help interpret the finding of the study more significantly. Authors can add the most recent work on correlation between YAP-1 and CRC.

---

## Round 0.2 · Minor Revisions

Please address or at least discuss Reviewer 2 comments.

Reviewer 1 ·

Basic reporting

No comments

Experimental design

No comments

Validity of the findings

No comments

Additional comments

Authors have taken a good-faith effort in addressing comments from the initial submission. I have now reviewed the changes they have made in the revised version and quite happy with the overall manuscript.

Reviewer 2 ·

Basic reporting

Raw data. The authors declined to provide raw data such as TNM stages, differentiation grade for each patient due to the concern for patient confidential information, although in principle information such as TNM stage should not be any different from the already provided DFS, DSS, Chemotherapy raw information. The authors provided the manual grading of H-scores before averaging. However, the nuclear H-score appear weird. All nuclear H-scores are in the increment of 5, which should be the case if it is calculated in the same way as the cytoplasmic H-score.

Experimental design

No comment

Validity of the findings

1. The authors said they had compared the group with chemotherapy (either high cytoplasmic YAP1 or low cytoplasmic YAP1) to the corresponding group without chemotherapy as suggested, and revised their conclusion based on this comparison to “Stage III CRC patients with high cytoplasmic YAP1 expression did not receive significant survival benefit from adjuvant chemotherapy”. However, they did not make any update to Figure 3 to reflect this comparison.
2. The authors rewrote their final conclusion of Figure 3 to “therefore, as compared to the patients in stage III with lower cytoplasmic YAP1 expression, those with high cytoplasmic YAP1 expression did not receive significant survival benefit from adjuvant chemotherapy”. This conclusion is exciting, but it is not supported by their data at all and is misleading. It sounds like for patient with lower cytoplasmic YAP1 expression, they did receive significant survival benefit from adjuvant chemotherapy; and for patient with high cytoplasmic expression, they did not. Related to the point above, the author did not show this comparison in their Figure 3. And based on the data presented in Figure 3, chemotherapy did not seem to benefit patient with lower cytoplasmic YAP1. Instead, chemotherapy even appeared to have an adverse effect on the survival of patients with low cytoplasmic YAP1, although this may largely be due to the small sample size of the group with low cytoplasmic YAP1 and without chemotherapy. Either way, the data does not support the conclusion drawn by the authors.

Additional comments

In this revised manuscript, Dong and colleagues addressed some of my questions. However, they did not address my concern on the adjuvant chemotherapy result, which is a particularly important part of this study. Their conclusion is not supported by the data presented. Please see specific comments for details.

Reviewer 3 ·

Basic reporting

The authors have incorporated my suggestions and answered all my queries.

Experimental design

No comments

Validity of the findings

No comments

---

## Round 0.3 · accepted · Accept

The revisions have been concluded properly.